# Anthropogenic CH₄ Emissions in the Yangtze River Delta Based on A "Top-Down" Method

**Wenjing Huang [1,2], Wei Xiao [1,2], Mi Zhang [1], Wei Wang [1], Jingzheng Xu [3], Yongbo Hu [1], Cheng Hu [1,4], Shoudong Liu [1] and Xuhui Lee [1,2,5,*]**

[1]   Yale-NUIST Center on Atmospheric Environment, Nanjing University of Information, Science and Technology, Nanjing 210044, China; 20181108059@nuist.edu.cn (W.H.); wei.xiao@nuist.edu.cn (W.X.); zhangm.80@nuist.edu.cn (M.Z.); wangw@nuist.edu.cn (W.W.); 20171103089@nuist.edu.cn (Y.H.); huxxx991@umn.edu (C.H.); lsd@nuist.edu.cn (S.L.)
[2]   NUIST-Wuxi Research Institute, Wuxi 214073, China
[3]   Radio Science Research Institute Inc., Wuxi 214073, China; xu.jingzheng@js1959.com
[4]   Department of Soil, Water, and Climate, University of Minnesota-Twin Cities, St. Paul, MN 55108, USA
[5]   School of Forestry and Environmental Studies, Yale University, New Haven, CT 06511, USA
*    Correspondence: Xuhui.lee@yale.edu (X.L.)

**Abstract:** There remains significant uncertainty in the estimation of anthropogenic CH₄ emissions at local and regional scales. We used atmospheric CH₄ and CO₂ concentration data to constrain the anthropogenic CH₄ emission in the Yangtze River Delta one of the most populated and economically important regions in China. The observation of atmospheric CH₄ and CO₂ concentration was carried out from May 2012 to April 2017 at a rural site. A tracer correlation method was used to estimate the anthropogenic CH₄ emission in this region, and compared this "top-down" estimate with that obtained with the IPCC inventory method. The annual growth rates of the atmospheric CO₂ and CH₄ mole fractions are $2.5 \pm 0.7$ ppm year$^{-1}$ and $9.5 \pm 4.7$ ppb year$^{-1}$, respectively, which are 9% and 53% higher than the values obtained at Waliguan (WLG) station. The average annual anthropogenic CH₄ emission is $4.37 \, (\pm 0.61) \times 10^9$ kg in the YRD (excluding rice cultivation). This "top-down" estimate is 20–70% greater than the estimate based on the IPCC method. We suggest that possible sources for the discrepancy include low biases in the IPCC calculation of emission from landfills, ruminants and the transport sector.

**Keywords:** "top-down" method; the Yangtze River Delta; CO₂; CH₄; annual growth rate; anthropogenic CH₄ emissions

## 1. Introduction

The source apportionment of CH₄ is important for the study of carbon cycle and climate change. The mole fraction of CH₄ in the atmosphere increased by 157% from 1750 to 2011 [1,2]. As the second largest greenhouse gas next to CO₂, CH₄ has a warming potential of 28 times that of CO₂ with a 100-year time horizon [2]. In addition to the greenhouse effect, CH₄ also affects the chemical and photochemical reactions in the atmosphere [3]. The annual growth rate of atmospheric CH₄ was $6.9 \pm 2.4$ ppb year$^{-1}$ from 2007 to 2017 [4]. However, the source contributions of CH₄ have so far not been accurately quantified, especially at the regional and the city scale [5].

Anthropogenic CH₄ emissions account for 50–65% of the global CH₄ emissions of $5.82 \, (\pm 0.5) \times 10^{11}$ kg year$^{-1}$ [6,7]. Large uncertainties still exist in regional anthropogenic emission estimates. These estimates are usually based on the Intergovernmental Panel on Climate Change (IPCC) inventory method. The IPCC method aggregates the CH₄ emissions generated by different anthropogenic

activities and sums up the individual components to the domain of interest. One problem is that activity data, such as landfill and livestock, and emission factors cannot be accurately determined at the regional and the city scale [8–10]. A study in Beijing found that the uncertainty caused by landfill accounts for nearly half of the total uncertainty in the $CH_4$ emission estimate [11]. Accurate and timely calculations of anthropogenic $CH_4$ emissions at the regional scale are necessary for assessing the effectiveness of emission reduction policies.

Anthropogenic greenhouse gas emissions can also be estimated from observations of the gaseous concentrations in the atmosphere ("top-down" methods). The "atmospheric method" used in this study is one of the "top-down" approaches. One reason for using the atmospheric method is that many sources of anthropogenic $CH_4$ cannot be quantified with traditional methods, such as the chamber method [12–14]. The atmospheric method requires simultaneous concentration measurements of the target gas ($CH_4$) and a tracer gas (usually $CO_2$) when there is no disturbance from sinks or other natural sources [15]. In an observational study using aircraft profile measurement over a broad region of Alaska and Canada, the concentrations of $CH_4$ and $CO_2$ increase synchronously with height, showing a strong positive correlation between the two gases [16]. A similar positive relationship also exists in the surface air in a moderately polluted urban atmosphere of Boulder, USA [16]. The explanation for the positive relationship is that the two gases share common source areas and undergo the same long-range transport [17,18]. The concentration ratio between the two gases were used to estimate the $CH_4$ emissions in the densely populated urban areas in Southern California, showing that inventory $CH_4$ emission estimates for these urban areas are lower in comparison to the "top-down" atmospheric estimate [19]. In a study of anthropogenic $CH_4$ emissions in the Los Angeles megacity, a ground-based remote sensing concentration measurement with 29 different surface targets was used to spatially resolve $CH_4$:$CO_2$ emission ratio, once again relying on the linear relationship between the two gaseous concentrations [20].

The atmospheric method is based on a strong correlation between the observed concentration values of two relatively inert gases $CH_4$ and $CO_2$. Because the lifetime of these two gases in the atmosphere (7–11 years and 50–200 years, respectively) [21] is much longer than hourly time scales at which the observations are made, the linear slope value of the regression is essentially equivalent to the ratio of their anthropogenic emission strengths. In the applications cited above, the $CH_4$ emission flux is computed as the product of the concentration regression slope and the anthropogenic $CO_2$ emission flux, the latter of which can be obtained reliably with the IPCC inventory method [22].

The atmospheric method has been used to track emissions of other gas species besides $CH_4$. This method was used to infer anthropogenic Hg emissions in Northeast USA, with wintertime Hg and $CO_2$ concentration data and $CO_2$ as the tracer [23]. Simultaneous observations of the $CO_2$ and CO concentrations in a suburban site outside Beijing was used to determine the efficiency of fossil fuel combustion [24]. The concentrations of $CO_2$ and CO observed in the Asian outflow air combined with a three-dimensional global chemical transport model were used to quantify $CO_2$ emissions in East Asia [25].

In this paper, we report long-term (five years), near-continuous, and simultaneous observations of atmospheric $CH_4$ and $CO_2$ at a lake site near Wuxi, Jiangsu Province, China. Our main objective was to quantify the $CH_4$ emission in the Yangtze River Delta (YRD) and its interannual variability, using the atmospheric tracer method described above. The second objective was to evaluate the validity of the $CH_4$ emissions calculated with the "bottom-up" IPCC inventory method against the atmospheric or "top-down" emission estimates.

## 2. Experiments

### 2.1. Study Area and Observational Site

The YRD in East China occupies only 2.1% of the land area of China (including Jiangsu Province, Zhejiang Province, Anhui Province and Shanghai), but contributes 1/4 of the total economic output [26].

The Wuxi Municipality is located roughly in the middle of the Yangtze River Delta, with a population of about 5 million. Other major cities in this region are Shanghai, Nanjing, Hangzhou, Ningbo and Hefei.

The observational site is located at the Taihu Lake Ecosystem Observatory of the Chinese Academy of Sciences (31.4197° N, 120.2139° E) in Wuxi. Measurements of the $CO_2$ and $CH_4$ concentrations were made on a platform about 200 m from the north shore of the lake (Figure 1). The platform was once also part of the Lake Taihu Eddy Flux Network (site id MLW) [27]. The site is surrounded by water, scattered farmlands and residential buildings. The closest traffic road is about 10 km away. The prevailing wind is northwesterly in the winter (Figure S1) and southerly in the summer. In the winter season, the landscape upwind is mostly rural (Figure S1).

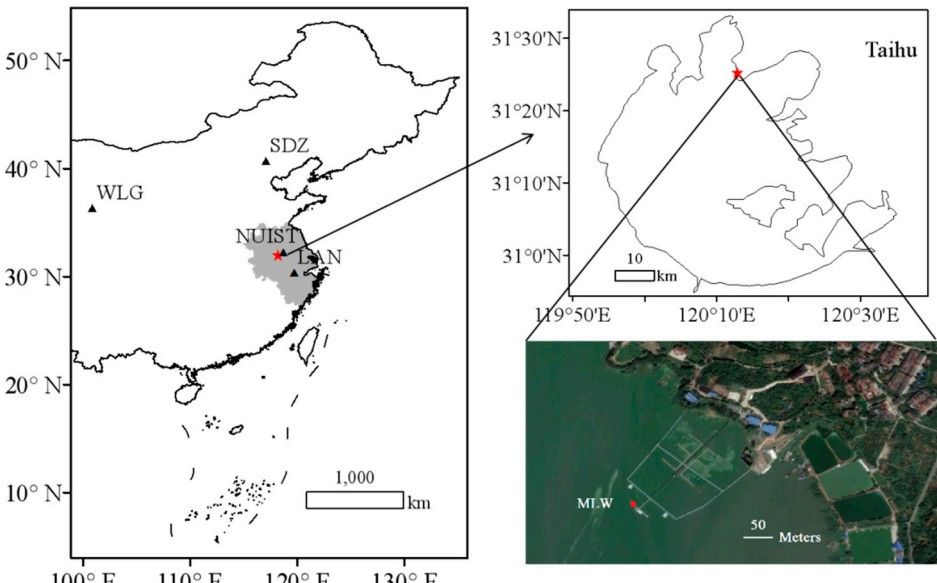

**Figure 1.** Location of the MLW site (pentagram in red color), Nanjing University of Information Science and Technology (NUIST) and three Chinese WMO/GAW stations, including Lin'an (LAN), Shangdianzi (SDZ) and Waliguang (WLG).

*2.2. Trace Gas Analyzer*

The analyzer we used was based on wavelength scanning cavity ring-down spectroscopy (model G1301 from 14 May 2012 to 6 July 2013 and model G2301 from 20 October 2013 until now, Picarro Inc., Sunnyvale, CA, USA). The measurement frequency is 1 Hz (sampling rate) and the precision (5-s mean) is 0.15 ppm for $CO_2$ and 1 ppb for $CH_4$. The air inlet is placed at a height of 3.5 m above the water surface. The water vapor concentration measured by the instrument was used to remove the water vapor dilution effect so the concentrations of $CH_4$ and $CO_2$ are expressed as molar ratio of $CH_4$ or $CO_2$ to dry air.

The observation period in this study was from May 2012 to April 2017. A large data gap occurred between July and October 2013 owing to instrument malfunction. In the second half of 2016, some data were lost because of loss of power on the platform.

The data from the analyzer were averaged to half-hour intervals. Standard deviation in half an hour and a five-point moving average method with a threshold of 1.5 times the standard deviation were used to filter the outliers. The daily average was achieved when daily data exceeded 75%.

For model G1301, the $CO_2$ and $CH_4$ measurements were calibrated twice (2 May 2012 and 22 June 2012) and the details can be found in the supplementary materials of a previously published paper [28]. Model G2301 was calibrated three times on 11 September 2013, 4 November 2015 and 25 August 2016, respectively. The mole fractions of $CO_2$ and $CH_4$ were calibrated against two standard $CO_2$ gases (concentration: 490 ppm or 491 ppm and 385 ppm, National Primary Standard prepared by the National Institute of Meteorology (NIM), China) and two standard $CH_4$ gases (2.02 ppm and 3.05 ppm

or 3.52 ppm, National Primary Standard) each time. The relative error is 0.62–0.37% for $CO_2$ and 0.81–0.41% for $CH_4$. We used a dew-point generator (model 610, LI-COR, Inc., Lincoln, NE, USA) to correct the analyzers humidity measurement. The humidity level of the air coming out of the dew-point generator was regulated at five levels and measurement at each humidity level lasted 15 min. The first several minutes when the measurement was transitional were excluded from the analysis. A linear regression fit generated from saturation vapor mixing ratio and observed vapor mixing ratio resulted in a slope value of 0.97–0.99.

No standards were available for us to trace our calibration gases to the WMO scale. NIM participated in two inter-agency comparison experiments on calibration standards including standards traceable to the WMO scale. The results can be found in references [29,30].

*2.3. The IPCC Inventory Calculation*

The IPCC emissions inventory is a "bottom-up" approach. It takes emission activity data from different economic sectors into account and multiplies them by the corresponding emission factors to estimate the emission. The activity data used in this study were obtained from China Energy Statistical Yearbook [31–34], China Statistical Yearbook [35–38], China Rural Statistical Yearbook [39–42], Jiangsu Statistical Bureau, Auhui Statistical Bureau, Zhejiang Statistical Bureau, Shanghai Statistical Bureau and Wuxi Statistical Bureau. The data on crop straw burning were derived from crop yields combined with grain-to-straw ratio [43]. The data on firewood usage from 1991 to 2006 were estimated with the ratio of firewood usage in the YRD combined with the total firewood usage [44]. Then a time-varying line of the firewood usage was fitted to estimate the 2012–2015 firewood usage in the YRD. We used the default emission factors provided by IPCC if no domestic values are available, such as industry energy consumption, industry processes and livestock. $CH_4$ emissions from landfills were based on the first order decay model provided by IPCC, taking into account local climatic conditions, the landfill waste volume, organic carbon content and waste age [11,45,46]. $CH_4$ emissions from rice paddies accounted for different varieties of rice acreage and the corresponding growth period [47–49]. The proportion of open-pit mining to the total mining volume has increased year by year in the YRD region [50]. The $CH_4$ fugitive emissions in the YRD mainly come from coal mining in Xuzhou in Jiangsu Province and Huainan, Huaibei, Fuyang, etc. in Anhui Province. The remaining emission factors for other sectors not listed here were obtained from the relevant literature [51,52].

The Monte Carlo method [52,53] was used to obtain uncertainty ranges of the inventory calculations. The uncertainties of the IPCC method arise from the choice of emission factors and from uncertainties in the activity data. These uncertainties were assumed to follow uniform distributions. The range of variations of the emission factors were given by IPCC or domestic values. For the activity data, an uncertainty range of 10% was assumed. A total of 400,000 ensemble members were calculated to determine a probability distribution function and estimate the emission uncertainties. A 95% confidence interval was used to quantify the random errors.

The $CO_2$ inventory is well quantified and with less uncertainty. For example, in Austria, Norway, the Netherlands, the UK and the USA, the uncertainty of $CO_2$ emission factors and activity data for main sources is between ± 0.5% and ± 7% of the means [54]. In comparison, the uncertainty of the $CH_4$ factors sector is ± 20% to ± 50% of the means [54]. The overall uncertainty of the $CO_2$ emission estimates is 2–4%, much smaller than that of $CH_4$ (17–48%) or $N_2O$ (34–230%) [54–56].

*2.4. Application of the Atmospheric Method*

We used a geometric mean regression to determine the slope of the $CH_4$ molar mixing ratio against the $CO_2$ molar mixing ratio. Because uncertainties exist in both $CO_2$ and $CH_4$ concentration measurements, geometric mean regression gives more robust parameter estimates than the ordinary least squares regression [57]. Moreover, there is slight variations between the four seasons according to the national level fuel consumption [58], so we focused on wintertime (December to February inclusive) measurements because plant photosynthesis is minimal and atmospheric $CO_2$ variations are driven

primarily by anthropogenic sources. The annual anthropogenic $CH_4$ emission flux is the product of the regression slope in winter and the anthropogenic $CO_2$ flux derived from the IPCC inventory method. We refer to this flux as the "top-down" estimate.

The uncertainty of the atmospheric method comes from two aspects. The first is a result of the regression procedure, and is characterized by the standard deviation of the regression estimate of the geometric mean slope. The second source of uncertainty is caused by the anthropogenic $CO_2$ emission calculation using the IPCC method. The overall uncertainty of the atmospheric $CH_4$ emission estimate was calculated with the Monte Carlo method that combines these two sources of uncertainty.

The daytime $CO_2$ and $CH_4$ concentrations are indicative of source contributions at the regional scale if there are no direct nearby emissions. This is because the daytime atmospheric boundary layer (ABL) is well mixed and there is no nearby direct emission disturbance. At a suburban site in Xiamen in southeastern China, the vertical profile of $CO_2$ concentration shows little variation with altitude (below 350 m) between 8:40 and 15:45 [59]. At the Zotino Tall Tower Observatory, $CO_2$ and $CH_4$ in the atmosphere becomes well-mixed and their concentrations become nearly indistinguishable at six height levels (4, 52, 92, 157, 227 and 301 m) during day in summer [60]. The ABL height at the MLW site in the winters of 2014–2016, simulated with the Meteoinfo open-source software [61], varies between 570 m and 970 m in the midday period (10:00–17:00; Figure 2). The Meteoinfo program used the Data Assimilation System (GDAS1) as input data, and the predicted ABL height was interpolated spatially to the MLW site. The analysis of Potential Source Contribution Function (PSCF) calculated with the HYSPLIT trajectory model with the 500-m height, the approximate mean height of the mid-point of the ABL, as the end point is shown in Figure 3. The PSCF value can be interpreted as the conditional probability for the specific grid cell, or the ratio of the number of endpoints falling in the grid cell with high concentration (>85th percentile concentration at the receptor site) to the total number of endpoints falling in the grid cell. Grid points having high PSCF values are likely to be potential source regions contributing to the observed concentrations [62]. A weighting function was introduced to deal with the bias brought by PSCF value when the number of endpoints falling in the grid cell was small. The choice of the function was set according to that reported by Sigler and Lee [63]. The calculation was performed for December 2014 with 48-h backward trajectories at 10:00, 13:00 and 16:00 LST each day. In Figure 3, the area with the PSCF value greater than the typical threshold of 0.1 [51,64] is $3.8 \times 10^5$ $km^2$, of which 76% fall in the political boundary of the YRD. As a result, the daytime observational data at the MLW site can represent the source signatures of the YRD, consistent with a similar study conducted in Nanjing [51].

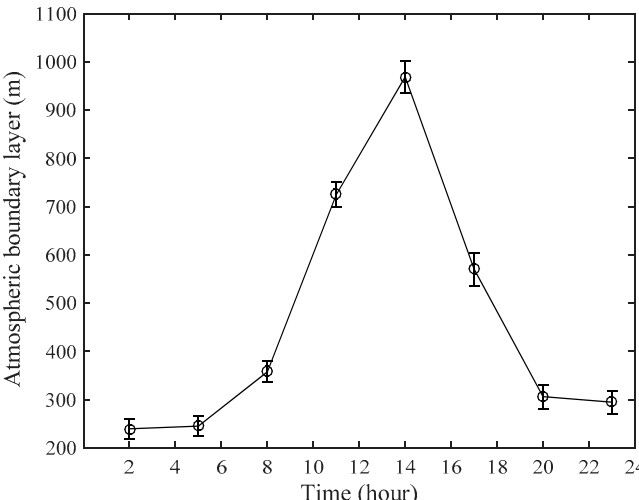

**Figure 2.** Diurnal variation of the boundary layer height at the MLW site. Error bars are $\pm$ 1 standard deviation of the mean.

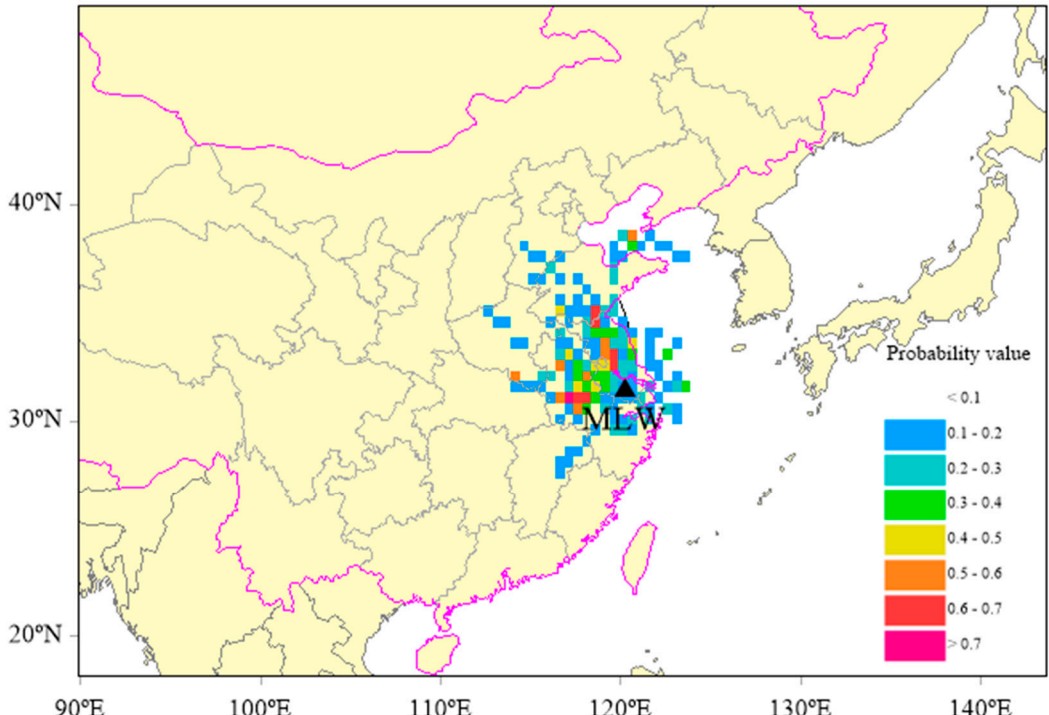

**Figure 3.** Spatial distribution of the potential source region contributions simulated for December 2014.

The footprint analysis revealed the source region mostly likely to have impacted the daytime measurement at MLW. The actual probability value, or weighting factor, was not used later when we aggregated the inventory emission data to the whole YRD region.

At night (23:00–05:00), an inversion layer typically prevails near the ground surface, with high atmospheric stability. The mean height of the boundary layer is 260 m (Figure 2). Because of the strong stability, the $CO_2$ and $CH_4$ emitted by anthropogenic sources are trapped near the surface. The lack of mixing implies that the source areas of the observed concentration may span only several kilometers [65]. In other words, the regression slope represents more the emission ratio of local sources than the emission ratio of regional sources, although the exact spatial representation of nighttime observations needs to be further studied.

## 3. Results

### 3.1. Temporal Variations of $CO_2$ and $CH_4$ Concentrations

Figure 4 shows the temporal variations of half-hourly atmospheric $CH_4$ and $CO_2$ mole fractions during the observation period. The data show significant periodic fluctuations through the 24-h cycle, especially for $CO_2$. Data for 2014 and 2015 are nearly gap-free and are representative of seasonal variations.

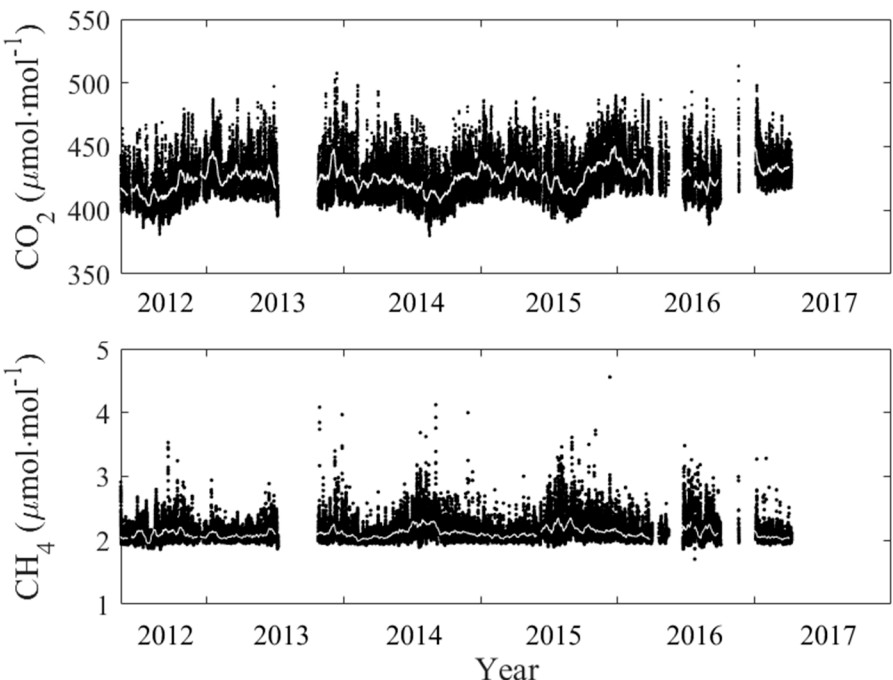

**Figure 4.** The half-hourly $CO_2$ and $CH_4$ mole fractions during the observation period (black dots). The white points are 14-day moving averages.

Figure 5 shows the diurnal composite concentrations for each of the four seasons in 2014. The diurnal variations of $CO_2$ and $CH_4$ show similar patterns in all seasons. In the spring (March, April and May) and autumn (June, July and August), the peaks appear at about 07:00 and the troughs appear at around 17:00. In the summer, the peaks occur at 04:00 and the minimum still appear around 17:00. In the winter (December, and January and February in the next year), the diurnal variations are gentler than in the other three seasons.

Figure 6 shows a comparison of our monthly mean $CO_2$ and $CH_4$ concentrations with those observed at Nanjing University of Information Science and Technology (NUIST; 32.20° N, 118.72° E), about 170 km to the northwest of the MLW site. The concentrations observed during the same time period at Mt. Waliguan (WLG, 36.28° N, 100.90° E, 3810 m above the mean sea level), a WMO baseline station representing the background atmosphere for Asia, are also shown [66,67]. NUIST is located at the outskirt of Nanjing, surrounded by residential areas and traffic roads and in the vicinity of two industrial complexes [68]. The atmospheric $CO_2$ molar fraction is highest at NUIST, followed by MLW and lowest at WLG. Among the three sites, the strongest seasonality of the atmospheric $CO_2$ molar fraction occurs at the MLW site, with low values in the summer and high values in the winter. The atmospheric $CH_4$ molar fraction at the MLW site shows an opposite seasonality to $CO_2$, with high values in the summer and low values in the winter.

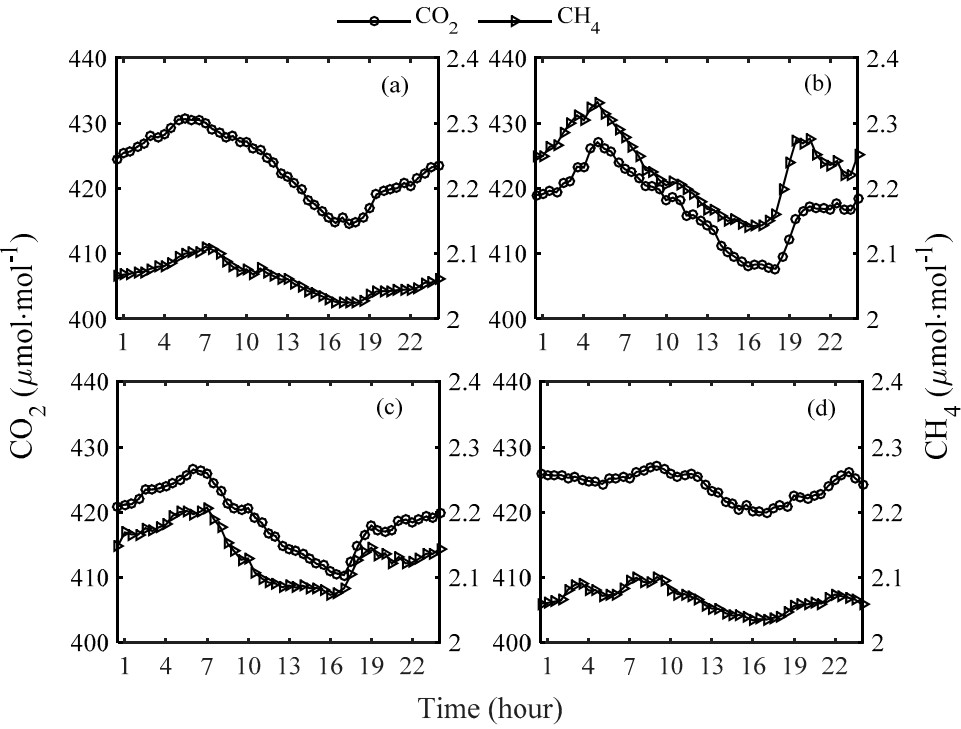

**Figure 5.** Diurnal variations of the molar fraction of $CH_4$ and $CO_2$ in the atmosphere in the four seasons in 2014: (**a**) spring; (**b**) summer; (**c**) autumn; and (**d**) winter.

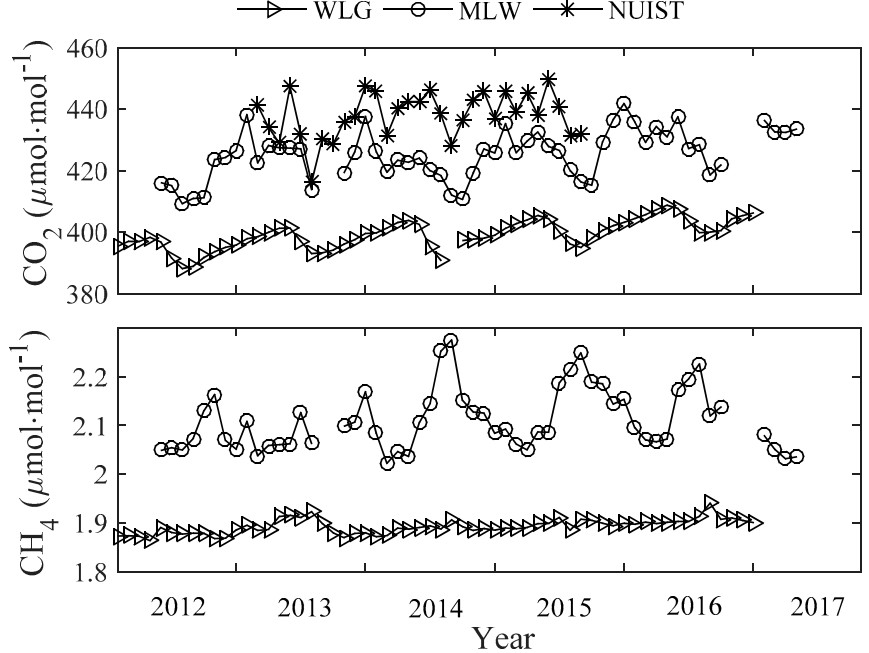

**Figure 6.** Variations of monthly molar fraction of $CO_2$ in the atmosphere at Nanjing University of Information Science and Technology (NUIST), WLG and MLW and monthly molar fraction of $CH_4$ at WLG and MLW from 2012 to 2017.

The monthly time series were deseasonalized to avoid the interference of seasonality of $CH_4$ and $CO_2$. A relative anomaly of $CH_4$ and $CO_2$ in a particular month of a given year was computed as the difference between the given concentration and the average of all years in that month, divided by the standard deviation of all the concentrations during the research period for that specific month.

Then, the least square method was used to obtain the growth rates of $CO_2$ and $CH_4$. At the MLW site, the growth rates are $2.5 \pm 0.7$ ppm year$^{-1}$ for $CO_2$ and $9.5 \pm 4.7$ ppb year$^{-1}$ for $CH_4$. These rates are higher than those at WLG ($2.3 \pm 0.2$ ppm year$^{-1}$ for $CO_2$ and $6.2 \pm 1.7$ ppb year$^{-1}$ for $CH_4$).

### 3.2. Diurnal and Inter-Annual Variations of the $CH_4$ Versus $CO_2$ Regression Slope

Figure 7 shows the diurnal variations of the $CH_4$ versus $CO_2$ regression slope and their linear correlation coefficients (R) for the summer and the winter in 2015. The slope value was determined for each half-hour period of the day, using all the data collected in the same half-hourly period. In the summer, the regression slope is marked by a sudden rise at around 18:00, coinciding with the onset of the surface inversion layer. The summertime R is high at night but has very low values in the afternoon. In the winter, the slope value is generally smaller than the summer value, and its diurnal variation is weak. The wintertime R fluctuates around 0.8. The fact that the regression slope is stable and that the R value is high in the winter is not a surprise. In the winter months, biological sources of $CO_2$ and $CH_4$ are much weaker than the anthropogenic sources. For this reason, the data in winter were selected for further study.

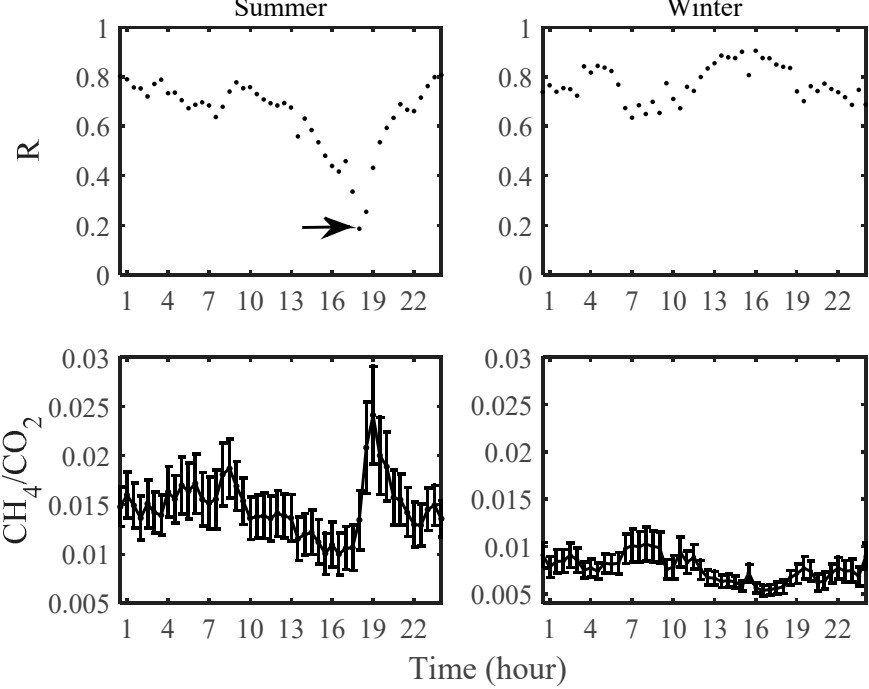

**Figure 7.** Diurnal variation of $CH_4$:$CO_2$ emission ratio in the summer (June, July, and August) in 2014 and in the winter (December, January and February) during 2014–2015. The arrow indicates the linear correlation at the *p* value of 0.05.

An important biological $CH_4$ source is wetland, including natural wetlands (offshore, coastal, swamp, lakes and rivers) and constructed wetlands. The total wetland area in the YRD is $5.4 \times 10^4$ km$^2$ (2013 value) [36], or 15% of the total area of the YRD. However, wintertime wetland $CH_4$ flux is generally weak. A study found that the methane emission rate in a freshwater wetland in Australia is less than 0.01 mmol m$^{-2}$ h$^{-1}$ in the winter, which is much lower than that in the summer (1.3–3.3 mmol m$^{-2}$ h$^{-1}$) [69]. Similarly, $CH_4$ emissions from natural wetlands in the YRD are $1.98 \times 10^8$ kg [51], far less than anthropogenic emissions in winter ($3.10 \times 10^9$ kg, Section 3.3). The Xixi wetland, one of the four major wetlands in the YRD, even becomes a weak sink of $CH_4$ in the winter (0.0019 mg m$^{-2}$ h$^{-1}$) [70]. Thus, $CH_4$ emissions from wetlands could be omitted when we compared the "top-down" and "bottom-up" estimates using wintertime observations.

Rice paddies are another important biological $CH_4$ source, contributing about 3.3–7.0% ($18.3 \times 10^9$ kg year$^{-1}$ to $8.8 \times 10^9$ kg year$^{-1}$, 1901–2010) [71] of the global $CH_4$ emissions ($5.58 \times 10^{11}$ kg year$^{-1}$, 2003–2012) [72]. China accounts for 20% of the rice production area in the world and the rice planting area in the YRD accounts for 18% of the China's total [35–38]. The important conditions for $CH_4$ production are organic matter applied (such as rice straw) and anoxic soils established in flooded paddies. As a typical monsoon climate zone in southeastern China, the growth period for rice is from May to October [73]. During non-rice growing season in the winter, $CH_4$ emissions from non-permanently flooded rice paddies are about 4–6% of the emissions in the growth season [74], and were ignored in the comparison with our "top-down" estimation.

Figures 8 and 9 are regression results using winter time observations made during 2012–2016. The regression was done separately for daytime (10:00–17:00 Beijing time) and nighttime (23:00–05:00 Beijing time) periods for consideration of different mixing conditions and source regions during the day and at night. In this regression, each data sample is a daytime or nighttime mean value. Taking the maximum drift over 24 h in $CO_2$ and $CH_4$ values (<120 ppb for $CO_2$ and <1 ppb for $CH_4$) for each point caused by the observational instrument into consideration, the slope was affected by <0.0001 ppm:ppm, which can be ignored comparing with the standard error caused by the fitting method (0.0005 in 2013). The daytime regression slope fluctuates between $0.0055 \pm 0.0006$ and $0.0068 \pm 0.0005$ ppm $CH_4$ per ppm $CO_2$ without an obvious interannual trend. The correlation coefficient R is greater than 0.8, and all four winters passed the 0.01 significance test.

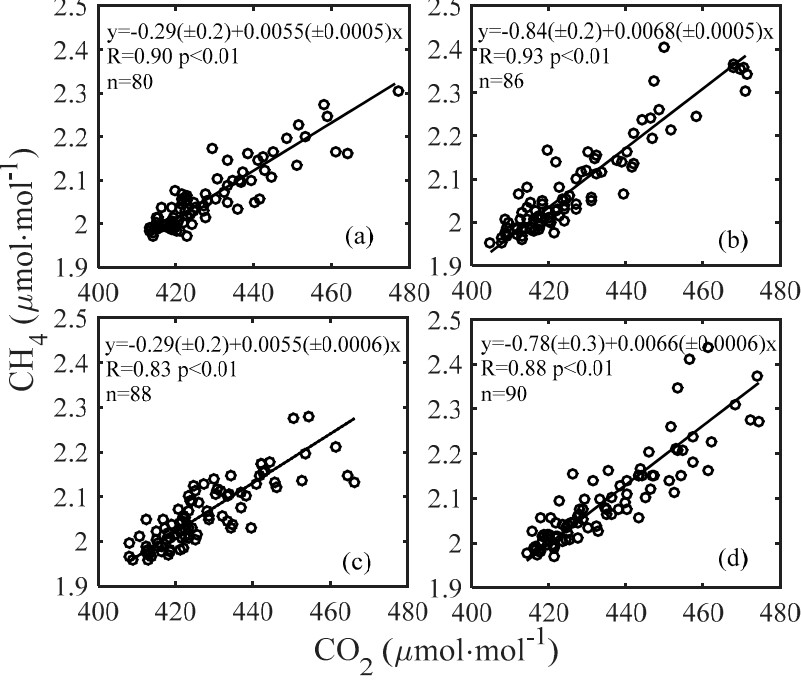

**Figure 8.** Scatter plots of winter (December–February) daytime $CH_4$ and $CO_2$ concentrations at MLW from 2012 to 2016. Each data point represents one daytime mean value. Regression statistics (regression equation, linear correlation R and number of samples) are also shown. Parameter ranges in the parentheses are 95% confidence bounds: (**a**) December 2012–February 2013; (**b**) December 2013–February 2014; (**c**) December 2014–February 2015; and (**d**) December 2015–February 2016.

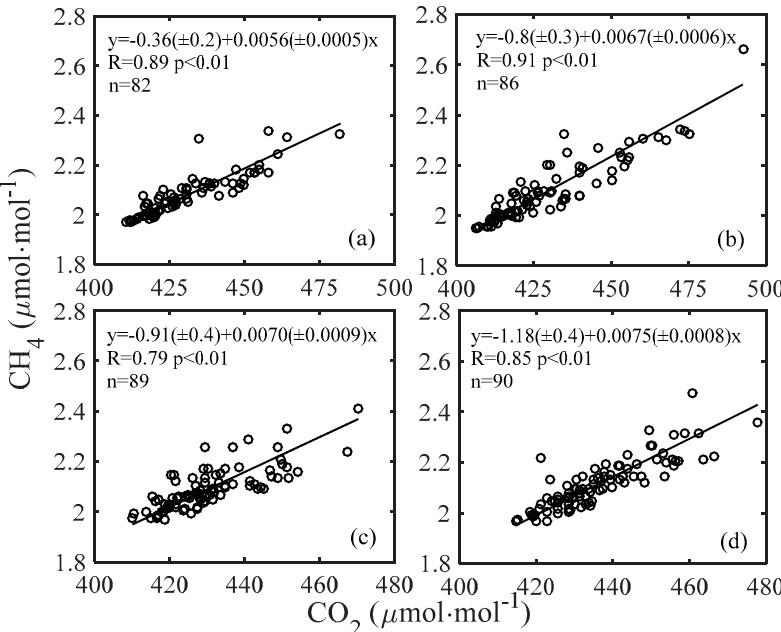

**Figure 9.** Same as Figure 8 except for winter (December–February) nighttime: (**a**) December 2012–February 2013; (**b**) December 2013–February 2014; (**c**)December 2014–February 2015; and (**d**) December 2015–February 2016.

The nighttime slope value increases from $0.0056 \pm 0.0005$ ppm/ppm for the winter of 2012–2013 to $0.0075 \pm 0.0008$ ppm/ppm for the winter of 2015–2016. With the exception of winter 2013–2014, the nighttime slope values are greater than the daytime values, implying more $CH_4$ emission per mole of $CO_2$ release by local anthropogenic sources than by regional sources. The correlation also passed the 0.01 significance test but the R values are slightly lower than the daytime R values.

An alternative approach to obtain the emissions ratio is to divide the $CH_4$ concentration enhancement over a background value by the $CO_2$ concentration enhancement. Here, we defined the clean background as the concentration at the 5th percentile. The emissions ratio, calculated as $(CH_{4, mean} - CH_{4, 5\%})/(CO_{2, mean} - CO_{2, 5\%})$ for the winter, is in good agreement with the slope of the $CH_4$ concentration versus $CO_2$ concentration (Figure S2).

In this study, we assumed that the daytime observations were influenced by sources located in the YRD. We used the EDGARv4.3.2 inventory data (2012) to understand the sensitivity of the emissions ratio to the spatial footprint. We found that by expanding the source region by 100 km from all sides of the YRD boundary, the $CH_4$:$CO_2$ emissions ratio changed by less than 1%.

### 3.3. Inventory Results

Table 1 shows the results of the $CO_2$ emission in the YRD in 2012. Industrial energy consumption is the largest emitter, followed by industrial processes, and the residential sector is the smallest emitter. The emission estimate for the transportation sector has the largest relative uncertainty, mainly due to the lack of accurate data on the annual driving range and the emission factor for different vehicle types and driving conditions [75]. The total emission amount was $19.18 \times 10^{11}$ kg in 2012. The Monte Carlo method gives an overall uncertainty of 10%. For comparison the total emission amount in 2009 was $15.35 \times 10^{11}$ kg [51].

**Table 1.** Anthropogenic $CO_2$ emissions in the YRD in 2012.

| Sector | Emission ($\times 10^{11}$ kg) | Percent of Total (%) |
|---|---|---|
| Industrial energy consumption [1] | 13.03 ($\pm$ 11%) | 67.9 |
| Industrial processes | 4.40 ($\pm$ 10%) | 23.0 |
| Transportation | 1.35 ($\pm$ 18%) | 7.0 |
| Household | 0.40 ($\pm$ 8%) | 2.1 |
| Total | 19.18 ($\pm$ 10%) | 100 |

[1] $CO_2$ emissions in manufacturing, commerce, and construction are also included in this sector.

The annual $CO_2$ emission for 2012–2015 is shown in Figure 10. The growth rate was 3% from 2012 to 2013 and decreased to 0.5% from 2014 to 2015. The reduced growth was mainly caused by the decrease in the industrial energy emission.

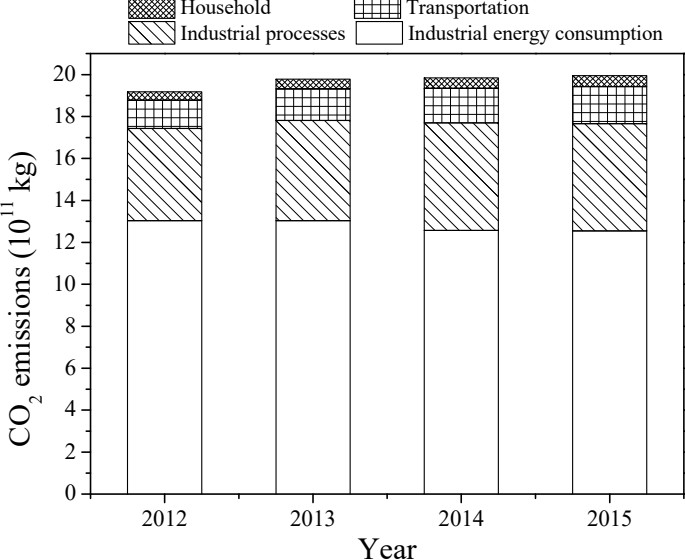

**Figure 10.** Annual anthropogenic $CO_2$ emissions from different sectors in the YRD from 2012 to 2015, based on the IPCC inventory method.

Similarly, Table 2 shows the anthropogenic $CH_4$ emissions in the YRD in 2012. Rice cultivation and coal mining are the major sources of anthropogenic $CH_4$ emissions. Emission estimates for landfills, wastewater treatment and fuel burning (traffic) emissions have large uncertainties. The total regional emission was $5.78 \times 10^9$ kg in 2012, increasing by 9.7% compared to that reported for 2009 [51]. The uncertainty of the total estimate (21%) is much greater than the uncertainty (10%) of the $CO_2$ emission estimate (Table 1), supporting the use of $CO_2$ as a tracer gas to calculate the $CH_4$ emission with the atmospheric method.

**Table 2.** Anthropogenic $CH_4$ emissions in the Yangtze River Delta in 2012, based on the IPCC inventory method.

| Sector | Emission ($\times 10^9$ kg) | Percent of Total (%) |
|---|---|---|
| Rice cultivation | 2.68 ($\pm$ 12%) | 46.3 |
| Landfill | 0.50 ($\pm$ 35%) | 8.7 |
| Wastewater treatment | 0.28 ($\pm$ 40%) | 4.8 |
| Livestock | 0.31 ($\pm$ 14%) | 5.4 |
| Fuel and Biomass burning | 0.32 ($\pm$ 17%) | 5.6 |
| Coal mining | 1.69 ($\pm$ 30%) | 29.2 |
| Total | 5.78 ($\pm$ 21%) | 100 |

Figure 11 shows the annual anthropogenic $CH_4$ emission in the YRD from 2012 to 2015. There was a slight downward trend during the study period, but there was a slight increase from 2014 to 2015.

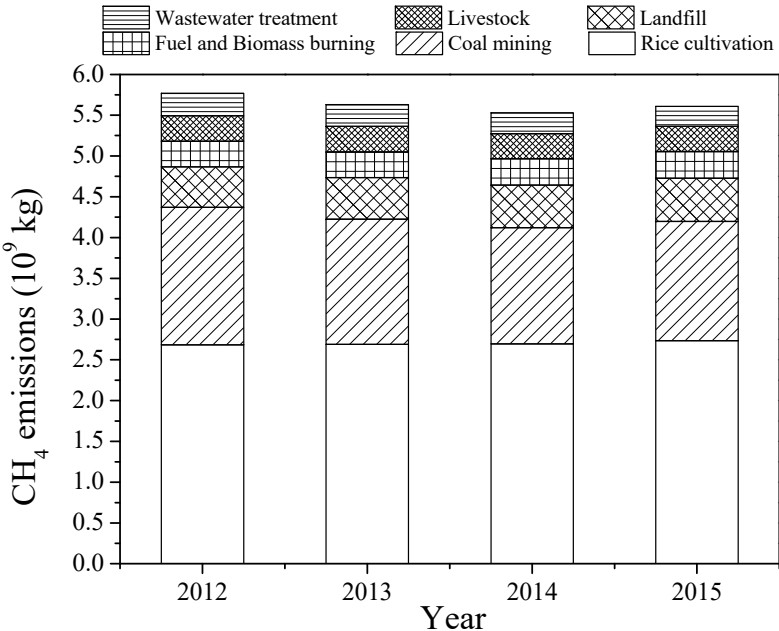

**Figure 11.** Annual anthropogenic $CH_4$ emissions from different sectors in the YRD from 2012 to 2015, based on the IPCC inventory method.

## 3.4. Comparison of $CH_4$ Emission Estimates between the Methods

Figure 12 shows the comparison of the annual anthropogenic $CH_4$ emissions obtained with the two methods. In this comparison, emissions from rice cultivation were excluded from the IPCC estimate because no rice is grown in the winter months. The "top-down" atmospheric estimate fluctuates year by year, in the range from $3.84 \times 10^9$ kg to $4.89 \times 10^9$ kg, which is 1.2–1.7 times the IPCC result.

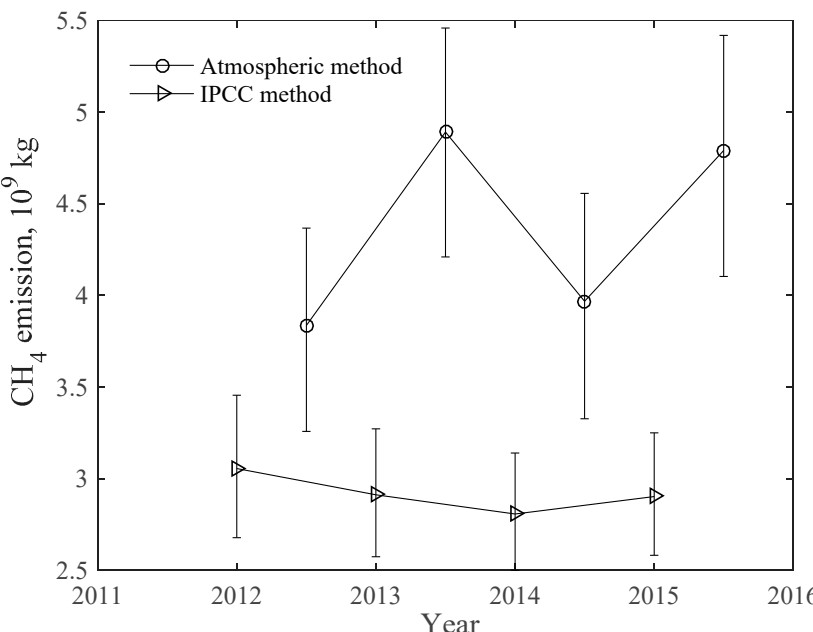

**Figure 12.** Annual anthropogenic $CH_4$ emissions (excluding rice cultivation) in the YRD from 2012 to 2015. Error bars indicate the uncertainty at a confidence level of 97.5%.

## 4. Discussion

### 4.1. Annual Growth Rates of $CH_4$ and $CO_2$ Concentrations

The observation lasted nearly five years. Due to the lack of data in some months, the annual average is not representative, thus the linear increase rate of monthly averages was used to characterize the annual growth rate. The $CO_2$ growth rate at the MLW site from 2012 to 2017 was $2.5 \pm 0.7$ ppm year$^{-1}$, which is 9% higher than observed at the WLG background station ($2.3 \pm 0.2$ ppm year$^{-1}$). A similar result was obtained from observations near Beijing, showing a 24% higher growth rate than at WLG [76]. The large growth rate is consistent with the anthropogenic $CO_2$ emissions trend in the YRD with an increase of 30% from $15.35 \times 10^{11}$ kg in 2009 [51] to $19.95 \times 10^{11}$ kg in 2015 (Figure 10). In this period, emissions from the transport sector increased by 88%, in line with the increasing car ownership, while emissions from industrial processes were nearly unchanged [51].

The annual growth rate of $CH_4$ at MLW ($9.5 \pm 4.7$ ppb year$^{-1}$) is nearly twice that at WLG ($6.2 \pm 1.7$ ppb year$^{-1}$). The increasing trend at MLW is close to that in the Lin'an station in Zhejiang Province, which is also located in the center of YRD, with a growth rate of $8.0 \pm 1.2$ ppb year$^{-1}$ from 2009 to 2011 [77]. The growth rate at WLG from 2012 to 2016 was quite similar to global mean growth rate (about 6 ppb year$^{-1}$, 2010–2014) [78,79]. The annual trend of $CH_4$ is 4.8 ppb year$^{-1}$ (2008–2013) at the Zhongshan station in Antarctica, which is the least influenced by human activities [80,81]. At the Shangdianzi regional background station in Beijing, atmospheric $CH_4$ concentration influenced by airmasses passing through the highly developed Beijing Municipality, Tianjin Municipality and Hebei Province increased at a rate of $10 \pm 0.1$ ppb year$^{-1}$ from 2009 to 2013, whereas atmospheric $CH_4$ concentrations influenced by airmasses originating from Russia, Mongolia, and the Inner Mongolia Autonomous Region of China increased at a rate of $6 \pm 0.1$ ppb year$^{-1}$ over the same period [82].

One reason for the high $CH_4$ growth rate at the MLW site is the steady increase in anthropogenic emissions according to the inventory data, although the "top-down" estimate does not reveal such a trend in anthropogenic emissions. Another reason may be related to the expansion of wetland areas. According to two national surveys of wetland resources, the total wetland area in the YRD obtained by the second census (2009–2013) increased by 54% in comparison with the first (1995–2003), from $3.5 \times 10^4$ km$^2$ to $5.4 \times 10^4$ km$^2$ [36]. About 67% of this growth is contributed by constructed wetlands, which increased by nearly 700%. In natural wetlands around the world, the $CH_4$ emission flux, ranging from $7.0 \times 10^3$ to $2.8 \times 10^4$ kg km$^{-2}$ year$^{-1}$ with an average of $2.1 \times 10^4$ kg km$^{-2}$ year$^{-1}$ [83,84] is lower than in constructed wetlands (ranging from $1.3 \times 10^3$ to $1.5 \times 10^5$ kg km$^{-2}$ year$^{-1}$, with an average of $4.7 \times 10^4$ kg km$^{-2}$ year$^{-1}$) [85]. The annual total emissions by wetlands for the two time periods (1995–2003 and 2009–2013) aere $1.89 \times 10^8$ kg year$^{-1}$ (range $1.6 \times 10^8$ to $14.8 \times 10^8$ kg year$^{-1}$) and $2.98 \times 10^8$ kg year$^{-1}$ (range $2.5 \times 10^8$ to $23.3 \times 10^8$ kg year$^{-1}$), respectively, using the Tier 1 approach provided by 2006 IPCC guidelines for moist and warm climates [8]. This method only considers diffusive emissions for natural wetlands during ice-free periods. Emissions from wetlands in two national surveys were 6.1% and 9.6% anthropogenic emission (excluding rice cultivation) in 2012 ($3.1 (\pm 0.5) \times 10^9$ kg, Table 2).

### 4.2. Comparison of the $CH_4/CO_2$ Emissions Ratio

Some sources, such as transportation and landfills, release both $CH_4$ and $CO_2$, but their $CH_4/CO_2$ emissions ratios are very different. The average $CH_4/CO_2$ emissions ratio of vehicle traffic is $4.6 (\pm 0.2) \times 10^{-5}$ ppm/ppm according to observations in traffic tunnels in Switzerland [86], indicating that $CH_4$ emissions from traffic emissions account for only a small fraction of the anthropogenic emission. In a study of eight cities in China, the traffic emissions ratio is $7.0 (\pm 3.6) \times 10^{-3}$ ppm/ppm [87]. The ratio is much larger than that in Switzerland. In comparison, $CH_4$ comprises up to 61% (median: 34%) of the total volume of landfill gas, with the remaining 3% to 69% (median: 33%) being $CO_2$ [88,89]. In the case of wetlands, the emissions ratio varies with the type of wetlands and sometimes is negative

because they act as sources of $CH_4$ but sinks of $CO_2$ [90–92]. The emissions ratio estimated from the regression slope is a composite signal of all the sources at the regional scale.

Table S1 summarizes the $CH_4/CO_2$ emissions ratio observed in different parts of the world. All these values were obtained as the regression slope of atmospheric $CH_4$ concentration against the $CO_2$ concentration. Some of these slope values were given by the cited literature, while others were estimated from the original concentration data. Our emissions ratio is close to those observed in Los Angeles and Pasadena, USA, but is higher than that observed in the daytime in Nanjing, China. The highest values occur in northern latitude remote sites far away from major industrial activities, such as Barrow, USA and Alert, Canada, due to abundant wetlands at these latitudes (>50% land area) [93]. About 60% of the total $CH_4$ emission from natural wetlands come from those between latitude 50° N to 70° N [83].

### 4.3. Comparison between the IPCC Method and the Atmospheric Method

Shen et al. measured atmospheric $CO_2$ and $CH_4$ molar fraction at a suburban site in Nanjing from June 2010 to April 2011 [51]. They argued that the regression slope from daytime data represents emissions ratio of sources in the YRD, and the regression slope from nighttime data represents emissions ratio of the local sources in the Nanjing Municipality. Their atmospheric estimate for $CH_4$ emission for sources in Nanjing is 200% higher than the IPCC estimate [51]. If we assume that the nighttime data in this study are also indicative of local sources in the Wuxi Municipality, our atmospheric estimate of the $CH_4$ emission in the winter would be $1.8 \times 10^8$–$2.3 \times 10^8$ kg, which is 4.4–5.7 times as large as the value calculated with the IPCC method (after exclusion of rice paddy emissions; Table 3). The IPCC method is far more uncertain at the urban scale than at the regional scale, for several reasons. First, cities do not have clear boundaries and direct emissions calculated based on urban statistics ignore some emissions of cities, such as aviation and waterways, so the total emission will be underestimated [94]. As the spatial scale increases, the dependence of the accounting area on cross-boundary transport of energy and material is smaller, and errors due to indirect emissions are reduced [95]. Second, emission factors at the cities are not measured accurately. Third, wastewater treatment and landfill are the two largest source categories in Wuxi, accounting for 45.6% of the city's total $CH_4$ emissions; these two sources have the largest uncertainty.

**Table 3.** Anthropogenic $CH_4$ emissions in Wuxi in 2012 based on the IPCC inventory method.

| Sector | Emission ($\times 10^7$ kg) | Percent of Total (%) |
|---|---|---|
| Rice cultivation | 3.05 ($\pm$ 13%) | 42.8 |
| Landfill | 1.81 ($\pm$ 38%) | 25.4 |
| Wastewater treatment | 1.45 ($\pm$ 40%) | 20.3 |
| Livestock | 0.48 ($\pm$ 22%) | 6.7 |
| Fuel and Biomass burning | 0.34 ($\pm$ 21%) | 4.8 |
| Coal mining | — | — |
| Total | 7.13 ($\pm$ 26%) | 100 |

In comparison with the IPCC estimate for the YRD in the winter, the "top-down" atmospheric estimate in this study is 1.2–1.7 times the IPCC result. Shen et al.'s estimate of $CH_4$ emission from the atmospheric method is 20% lower [51] than the IPCC estimate, even though their measurement and our measurement both took place in the same region (YRD) and not too far from each other (170 km apart). One possible reason for the difference is that Shen et al.'s observation site is not far away from industrial complexes and traffic roads. Traffic $CH_4/CO_2$ emissions ratio is generally much lower than regional emissions ratios, as noted above. According to the "bottom-up" results provided by the Emissions Database for Global Atmospheric Research (EDGAR), the global $CH_4/CO_2$ emissions ratios in the chemical production sector, the metal production sector, and the public electricity and heat production sector were $1.2 \times 10^{-3}$, $6.4 \times 10^{-4}$ and $7.6 \times 10^{-5}$ ppm/ppm, respectively in 2012 (http://edgar.jrc.ec.europa.eu/overview.php?v=432&SECURE=123) [96]. The situation is

a little different in China. The $CH_4/CO_2$ emissions ratio is bigger for the chemical production sector ($1.8 \times 10^{-3}$ ppm/ppm), and is smaller for the metal production sector ($4.7 \times 10^{-4}$ ppm/ppm) and the public electricity and heat production sector ($3.4 \times 10^{-5}$ ppb/ppm). Thus, the $CH_4/CO_2$ emissions ratio for industrial complexes should be lower than regional emissions ratios. Because our measurement was made at a rural system not directly impacted by local traffic and industrial emissions (Figure S1), our emissions ratio should be more representative than that reported in [51].

Another reason for the difference between our study and the study by Shen et al. [51] is different source areas between Nanjing and MLW. Hu et al. [97] simulated the atmospheric $CO_2$ concentration in Nanjing City using the WRF-STILT model. They found that the NUIST site is mainly affected by the central and eastern regions of Anhui Province and the central and western regions of Jiangsu Province, with greater concentration contribution weights than other regions in the YRD.

Previous studies have also reported that estimates of $CH_4$ emission based on the "top-down" atmospheric method are higher than the "bottom-up" inventory estimates. In the Los Angeles metropolitan area in California, USA, anthropogenic $CH_4$ emissions calculated from two "top-down" approaches are 1.3–1.8 and 1.2–1.6 times, respectively, of the "bottom-up" estimates [20,98]. The US EPA inventory and EDGAR, two "bottom-up" methods, are shown to underestimate $CH_4$ emissions in the United States by a factor of about 1.5 and 1.7, with the underestimation coming from two sectors: livestock and fossil fuel extraction and processing [99]. That "bottom-up" results are generally smaller indicate that there are unknown emission sources or underestimates of the emission capacity of known sources, such as landfills, coal mining and wastewater treatment. Some scholars have found that based on carbon isotope observations, $CH_4$ emissions from landfills and ruminants are underestimated by the IPCC method [100]. In China, the methane emission rate is reported to be eight times the IPCC emission estimate for natural gas vehicles [78]. If this result is taken into consideration, the anthropogenic $CH_4$ emissions in the YRD would increase by 2.4% in 2015. In the urban Boston area, USA, natural gas loss rate from transmission, distribution and end use was $2.7 \pm 0.6\%$ of the total delivered gas, which is more than twice the result of the emission inventory [101]. In 2016, the length of natural gas pipeline reached $5.51 \times 10^5$ km in China, although the $CH_4$ concentration measurements along urban street transects in eight Chinese cities show no evidence of pipeline leakage [87].

## 5. Conclusions

Continuous observation of atmospheric $CO_2$ and $CH_4$ mole fraction was made at the MLW station near Wuxi at Lake Taihu from May 2012 to April 2017. These measurements were combined with anthropogenic $CO_2$ emission data in a "top-down" method to obtain an estimate of the anthropogenic $CH_4$ emission in the YRD. For comparison, the $CH_4$ emission was also calculated with the IPCC inventory method. The key results are as follows:

(1) The growth rates of the $CO_2$ and $CH_4$ molar fractions at the MLW site were $2.5 \pm 0.7$ ppm year$^{-1}$ and $9.5 \pm 4.7$ ppb year$^{-1}$, respectively, which are 9% and 53% higher than that observed at WLG over the same period.

(2) To avoid the interference of biological sources, we used the wintertime $CO_2$ and $CH_4$ concentration data to obtain the $CH_4/CO_2$ emissions ratio. Results indicate that the emissions ratio fluctuates between $0.0055 \pm 0.0006$ ppm/ppm (winters of 2012–2013 and 2014–2015) and $0.0068 \pm 0.0005$ ppm/ppm (winter of 2013–2014). These ratios are similar to those observed in Los Angeles and Pasadena, USA.

(3) According to the "top-down" method, the annual average anthropogenic emission of $CH_4$ in the YRD from 2012 to 2015 is 4.37 ($\pm 0.61$) $\times 10^9$ kg year$^{-1}$ (excluding rice cultivation), which is 1.2–1.7 times the result from the IPCC inventory.

(4) The "top-down" method also suggests that at the local scale, the IPCC inventory estimate for anthropogenic $CH_4$ emission in the Wuxi municipality may be biased low by 4.4−5.7 times.

**Supplementary Materials:** Figure S1: Prevailing wind direction measured at the MLW site in winter (December 2015–February 2016). Figure S2: $CH_4/CO_2$ emissions ratio (ppm/ppm) estimated with different methods in

different regions, Table S1: Summary of $CH_4/CO_2$ emissions ratio (ppb/ppm) estimated with the atmospheric method found in the literature. Supplementary data file is available at: https://yncenter.sites.yale.edu/data-access.

**Author Contributions:** Data curation, W.H., J.X. and Y.H.; Formal analysis, W.H.; Investigation, W.H.; Methodology, W.H.; Resources, W.X., J.X., Y.H. and S.L.; Supervision, W.X., M.Z., W.W., C.H. and X.L.; Validation, C.H. and X.L.; Writing—original draft, W.H.; and Writing—review and editing, X.L.

**Funding:** This research was supported by internal grants from NUIST-Wuxi Research Institute.

**Acknowledgments:** We are grateful to all of the staff who work at the WMO/GAW stations in China for collecting the data, and to the Greenhouse Gases Research Laboratory of the China Meteorological Administration (CMA) for data analysis. We appreciate Ed Dlugokencky, Andy Crotwell, and Kirk Thoning of NOAA for their help and support of the research measurements.

**Conflicts of Interest:** The authors declare no conflict of interest.

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
