# Peer review of "Anthropogenic CH4 Emissions in the Yangtze River Delta Based on A “Top-Down” Method"

_atmosphere, doi:10.3390/atmos10040185_

Reviewer 1 Report

This study estimates anthropogenic CH4 emissions in the Yangtze River Delta (YRD) in China using in-situ measurements from one rural site. While the paper is generally well-written and easy to follow, and a 5-year measurement data have significant value for future studies, I am concerned about the analysis approach in estimating regional/local anthropogenic emissions without considering the background. Both CH4 and CO2 are long-lived greenhouse gases; as a result, they are generally well-mixed globally. This background signal contributes to the most part of the measured CH4 signal in ambient air (for example, the background CH4 level is about 1.9 ppm, and the levels measured by this study mostly fall between 1.95 to 2.4ppm). Thus, the background signals should be removed before estimating regional/local emissions. This can be done by concurrent measurements at an upwind location of the targeted area. Alternatively, the author can look carefully into their data and find ‘cleaner’ data (e.g. under high wind conditions/ influenced by cleaner sectors of the region) to use as backgrounds.  Overall, it is not convincing that the results presenting in this study can be representative of the YRD emissions only. In addition, I cannot find the description of data availability to the public. As a data reporting study, it is important to make data available with publication to ensure reproducibility.

Some detailed comments:

1.     Measurement scale and calibration: Line 122-125, how are the CO2 and CH4 standards traceable to the WMO CO2 and CH4 scales? Why use 3 or 3.5 ppm standards for CH4, which are much higher than the ambient CH4 level. In this case, the authors should demonstrate the linearity of instrument response to the measured CH4 levels. How are the relative error calculated? 0.81 to 0.41% (more than 10 ppb) seems high considering that the Picarro has ~ 1 ppb uncertainty.

2.     Some citations do not match the reference list, e.g. Line 37-38, citation 4 and 5.

3.     Line 34, when comparing warming potential of CH4 and CO2, a time scale is needed to be specified.

4.     Line 53-54, it is not true that atmospheric method to estimate CH4 emissions requires simultaneous measurements of CO2. A mass balance approach does not. This should be worded more carefully. Also, the correlations do not work for summer, when significantly different sources are contributing the CO2 and CH4 signals. The authors should clarify this here.

5.     Line 58, can’t find the citation 17 that related to Boulder, USA CH4 emissions.

6.     Line 58-59, it is very misleading when saying CO2 and CH4 share the same sources, which they actually don’t. The author should reword it. The positive relationship is a result of well-mixed regional/local anthropogenic emissions.

7.     Line 72-73, need citation for the reliability of IPCC method.

8.     Figure 1, unclear what the symbols in the figure mean.

9.     Line 160, missing citation.

10.  Line 172-173, the day time PBL helps with the mixing, but the measurement site also needs to be free from direct nearby emissions.

11.  Line 178, Metoinfom program? It is unclear where the ABL data are from. From direct measurements or models?

12.  Fig.7, unclear what the x-axis means in hour.

13.  Fig.8 and 9,  as stated above, the calculations of CH4/CO2 do not remove influence of background. Thus, one cannot be certain that the regional/local emissions are responsible for these ratios. In addition, 5-year of data are combined together. Are the background trends of CO2 and CH4 removed before combining different year of data together?

14.  Line 370-372, it is important to state that the ratio method is intent for anthropogenic emissions only.

15.  Line 438, the authors need to support the statement that their measurement site is a rural site without direct influence of nearby traffic and industrial emissions. How is the data variability look like?

Author Response

Response to Reviewer 1 Comments

Point 1: I am concerned about the analysis approach in estimating regional/local anthropogenic emissions without considering the background. Both CH4 and CO2 are long-lived greenhouse gases; as a result, they are generally well-mixed globally. This background signal contributes to the most part of the measured CH4 signal in ambient air (for example, the background CH4 level is about 1.9 ppm, and the levels measured by this study mostly fall between 1.95 to 2.4 ppm). Thus, the background signals should be removed before estimating regional/local emissions. This can be done by concurrent measurements at an upwind location of the targeted area. Alternatively, the author can look carefully into their data and find ‘cleaner’ data (e.g. under high wind conditions/ influenced by cleaner sectors of the region) to use as backgrounds.  

Response 1: Thank you for your constructive comment. Jaffe et al. (2005) [1] use the slopes of gas A vs gas B correlations as emission ratios under assumptions of (1) no losses of the substances during the transport, (2) constant source with fixed emission ratios, and (3) constant background concentration during the transport event. These assumptions can be extended to multiple sources along the trajectory of transport of gases A and B [2].

In response to the reviewer’s comment, we have also performed emissions ratio calculation as suggested, and found good agreement with the slope value:

An alternative approach to obtain the emissions ratio is to divide the CH4 concentration enhancement over a background value by the CO2 concentration enhancement. Here we defined the clean background as the concentration at the 5th percentile. The emissions ratio, calculated as (CH4mean-CH45%) / (CO2mean-CO25%) for the winter, is in good agreement with of the slope of the CH4 concentration versus CO2 concentration (Supplementary Figure S2).” (Lines 321 -325)

Point 2: Overall, it is not convincing that the results presenting in this study can be representative of the YRD emissions only.

Response 2: According the PSCF analysis, the terrestrial source areas mainly fall in the YRD (Figure 3). In response to this comment, we have used the EDGARv4.3.2 inventory data (2012) to understand the sensitivity of the emissions ratio to the spatial footprint:

“In this study, we assumed that the daytime observations were influenced by sources located in the YRD. We used the EDGARv4.3.2 inventory data (2012) to understand the sensitivity of the emissions ratio to the spatial footprint. We found that by expanding the source region by 100 km from all sides of the YRD boundary, the CH4:CO2 emissions ratio changed by less than 1%.” (Lines 326 -329) 

Point 3: In addition, I cannot find the description of data availability to the public. As a data reporting study, it is important to make data available with publication to ensure reproducibility.

Response 3: The data file is now available at https://yncenter.sites.yale.edu/data-access. This data site is noted in the paper. (Line 506)

Point 4: Measurement scale and calibration: Line 122-125, how are the CO2 and CH4 standards traceable to the WMO CO2 and CH4 scales? Why use 3 or 3.5 ppm standards for CH4, which are much higher than the ambient CH4 level. In this case, the authors should demonstrate the linearity of instrument response to the measured CH4 levels. How are the relative error calculated? 0.81 to 0.41% (more than 10 ppb) seems high considering that the Picarro has ~ 1 ppb uncertainty.

Response 4: The standard gases were prepared by the National Institute of Meteorology. The standard gases they produce are widely used in greenhouse gas research in China (see details in [3]). But unfortunately we did not have access to gas standards traceable the WMO scale. We used two standard gases to calibrate CH4, one of which was lower than the ambient CH4 level (2.02 ppm) and the other was higher than CH4 level (3.05 or 3.52 ppm), to bracket the ambient concentrations (Figure 4). The relative error is the ratio of the absolute difference between the standard gas value and the value measured by instrument to the standard gas value. According to manufacturer., the analyzer’s drift is no greater ~ 1 ppb over 24 hr. Our instrument drift was greater than this because the calibration was done over a longer interval (~ 12 months). Picarro instruments are quite linear in the range of ambient concentrations. Our experience is that two-point calibration is adequate. (e.g., Xu J, X Lee, W Xiao, C Cao, S Liu, X Wen, J Xu, Z Zhang, J Zhao (2017) Interpreting the 13C/12C ratio of carbon dioxide in an urban airshed in the Yangtze River Delta, China. Atmospheric Chemistry and Physics, 17: 3385-3399.)

Point 5: Some citations do not match the reference list, e.g. Line 37-38, citation 4 and 5.

Response 5: corrected.

Point 6: Line 34, when comparing warming potential of CH4 and CO2, a time scale is needed to be specified.

Response 6: The time scale is now noted. (Line 36)

Point 7: Line 53-54, it is not true that atmospheric method to estimate CH4 emissions requires simultaneous measurements of CO2. A mass balance approach does not. This should be worded more carefully. Also, the correlations do not work for summer, when significantly different sources are contributing the CO2 and CH4 signals. The authors should clarify this here.

Response 7: We have changed the text to:

Anthropogenic greenhouse gas emissions can also be estimated from observations of the gaseous concentrations in the atmosphere ("top-down" methods). The "atmospheric method" used in this paper is one of "top-down" approaches.” (Lines 50 - 52)

We did point out that the method could only be used in the winter:

“…so we focus on wintertime (December to February inclusive) measurements because plant photosynthesis is minimal and atmospheric CO2 variations are driven primarily by anthropogenic sources.” (Lines 174 -176)

Point 8: Line 58, can’t find the citation 17 that related to Boulder, USA CH4 emissions.

Response 8: citation added.

Point 9: Line 58-59, it is very misleading when saying CO2 and CH4 share the same sources, which they actually don’t. The author should reword it. The positive relationship is a result of well-mixed regional/local anthropogenic emissions.

Response 9: This sentence has been reworded as "…CO2 and CH4 share the same source areas". (Line 61)

Point 10: Line 72-73, need citation for the reliability of IPCC method.

Response 10: citation added.

Point 11: Figure 1, unclear what the symbols in the figure mean.

Response 11: Legends added

Point 12: Line 160, missing citation.

Response 12: corrected

Point 13: Line 172-173, the day time PBL helps with the mixing, but the measurement site also needs to be free from direct nearby emissions.

Response 13: this is now noted. (Line 187)

Point 14: Line 178, Metoinfo program? It is unclear where the ABL data are from. From direct measurements or models?

Response 14: The text has been rewritten as

“… simulated with the Meteoinfo open-source software [50] varies between 570 m and 970 m in the midday period (10:00–17:00; Figure 2). The Weteoinfo program used the Data Assimilation System (GDAS1) as input data, and the predicted ABL height was interpolated spatially to the MLW site.” (Lines 192 -195)

Point 15: Fig.7, unclear what the x-axis means in hour.

Response 15: corrected.

Point 16: Fig.8 and 9, 5-year of data are combined together. Are the background trends of CO2 and CH4 removed before combining different year of data together?

Response 16: As noted in the figure captions, each subplot was for one winter. We did not remove background trends. (please also refer to point 1.) 

Point 17: Line 370-372, it is important to state that the ratio method is intent for anthropogenic emissions only.

Response 17: clarified. (Line 394)

Point 18: Line 438, the authors need to support the statement that their measurement site is a rural site without direct influence of nearby traffic and industrial emissions.

Response 18: In response, we have added a map with wind rose for the winter months (Supplementary Figure S1), The main residential and commercial areas in Wuxi are located to the northeast of the MLW site and the prevailing wind direction in the winter is northwest.

References

[1] Jaffe D.; Prestbo E.; Swartzendruber P.; Weiss-Penziasa P.; Kato S.; Takami A.; Hatakeyama S.; Kajii Y. Export of atmospheric mercury from Asia. Atmospheric Environment 2005, 39, 3029-3038, doi: 10.1016/j.atmosenv.2005.01.030.

[2] Brunke E.-G.; Ebinghaus R.; Kock H. H.; Labuschagne C.; Slemr F. Emissions of mercury in southern Africa derived from long-term observations at Cape Point, South Africa. Atmospheric Chemistry and Physics 2012, 12, :7465-7474, doi: 10.5194/acp-12-7465-2012.

[3] Xu, J.; Lee, X.; Xiao, W.; Cao, C.; Liu, S.; Wen, X.; Xu, J.; Zhang, Z.; Zhao, J. Interpreting the 13C/12C ratio of carbon dioxide in an urban airshed in the Yangtze River Delta, China. Atmospheric Chemistry & Physics 2017, 17, 3385–3399, doi: 10.5194/acp-2016-349.

[4] Shen, S.; Dong, Y.; Wei, X.; Liu, S.; Lee, X. Constraining anthropogenic CH4 emissions in Nanjing and the Yangtze River Delta, China, using atmospheric CO2 and CH4 mixing ratios. Adv Atmos Sci 2014, 31, 1343–1352, doi: 10.1007/s00376-014-3231-3.

Reviewer 2 Report

This is a valuable contribution about the estimation of anthropogenic CH4 emissions from CH4 and CO2 concentrations, which may be a noticeable topic in wetlands, such as the Yantze River Delta. The research is focused on CH4 and CO2 concentrations and emissions to develop the “atmospheric” or “top-down” method, whose results are compared to those from the usual bottom-up IPCC inventory method. However, the analysis of relevant meteorological variables, such as the boundary layer height and the air parcel trajectories, is also considered. The paper is elaborated, well-structured, and merits to be published in Atmosphere after the inclusion of some minor changes. Section 2.4 is composed by four paragraphs, the first, l. 165-171, is devoted to the calculation of the annual anthropogenic CH4 emission flux, and must remain in this section together with the fourth paragraph, l. 206-211. Moreover, since the authors indicate that the geometric mean regression is used, a short comment about the advantages and/or disadvantages of this method against the usual linear regression would be appreciated. The second and third paragraphs of Section 2.4, l. 172-205, are devoted to the evolution of the boundary layer height at the measurement site and the PSCF calculated with air parcel trajectories ending at this site. Consequently, these paragraphs should be placed at the beginning of the Results in a new Section entitled “3.1 Influence of meteorology on CO2 and CH4 concentrations”, or with a similar title.

Minor remarks

L. 54. Introduce one space after “(CH4)”.

Figure 1. Symbols ± and ^- should be avoided. Moreover, NUIST and WLG stations, cited in the Results, and Lin’an and Shangdianzi stations, cited in the Discussion, should be placed with a symbol accompanied by the corresponding text to identify them.

Figure 7. Are the x-axis labels hours or half-hours? If they are half-hours, they should be changed to hours.

L. 291. What is “<0.00001”?< p="">

L. 363. “CH4” should be replaced by “CH4”.

L. 500-506. This section should be supressed, since no information is introduced.

L. 649-650. “Hg0” should be replaced by “Hg0” and “222Rn” should be replaced by “222Rn”.

L. 683. “1991 2012” should be replaced by “1991-2012”.

L. 719. “CO2” should be replaced by “CO2”.

L. 774. “222Radon” should be replaced by “222Radon”.

Author Response

Response to Reviewer 2 Comments

Point 1: Section 2.4 is composed by four paragraphs, the first, l. 165-171, is devoted to the calculation of the annual anthropogenic CH4 emission flux, and must remain in this section together with the fourth paragraph, l. 206-211.

Response 1: adjusted.

Point 2: Moreover, since the authors indicate that the geometric mean regression is used, a short comment about the advantages and/or disadvantages of this method against the usual linear regression would be appreciated.

Response 2: The text has been modified to:

We used a geometric mean regression to determine the slope of the CH4 molar mixing ratio against the CO2 molar mixing ratio. Because uncertainties exist in both CO2 and CH4 concentration measurements, geometric mean regression gives more robust parameter estimates than the ordinary least squares regression [1].” (Lines 171 -173)

Point 3: The second and third paragraphs of Section 2.4, l. 172-205, are devoted to the evolution of the boundary layer height at the measurement site and the PSCF calculated with air parcel trajectories ending at this site. Consequently, these paragraphs should be placed at the beginning of the Results in a new Section entitled “3.1 Influence of meteorology on CO2 and CH4 concentrations”, or with a similar title.

Response 3: Thank you for your suggestion. We feel that the structure we have is more logical because the second and third paragraphs explain the conditions for using the atmospheric method and therefore should belong to the methods section.  

Point 4: L. 54. Introduce one space after “(CH4)”.

Response 4: corrected.

Point 5: Figure 1. Symbols ± and ^- should be avoided. Moreover, NUIST and WLG stations, cited in the Results, and Lin’an and Shangdianzi stations, cited in the Discussion, should be placed with a symbol accompanied by the corresponding text to identify them.

Response 5: done.

Point 6: Figure 7. Are the x-axis labels hours or half-hours? If they are half-hours, they should be changed to hours.

Response 6: changed.

Point 7: L. 291. What is “<0.00001”?< span="">

Response 7: The text has been clarified as:

Taking the maximum drift over 24 h in CO2 and CH4 values (<< span=""> 120 ppb for CO2 and << span=""> 1 ppb for CH4) for each point causing by the observational instrument into consideration, the slope will be affected by << span=""> 0.0001 ppm:ppm.” (Lines 301-304)

Point 8: L. 363. “CH4” should be replaced by “CH4”.

Response 8: done.

Point 9: L. 500-506. This section should be supressed, since no information is introduced.

Response 9: the text is now removed.

Point 10: L. 649-650. “Hg0” should be replaced by “Hg0” and “222Rn” should be replaced by “222Rn”.

Response 10: corrected.

Point 11: L. 683. “1991 2012” should be replaced by “1991-2012”.

Response 11: corrected.

Point 12: L. 719. “CO2” should be replaced by “CO2”.

Response 12: corrected.

Point 13: L. 774. “222Radon” should be replaced by “222Radon”.

Response 13: corrected.

Reference

[1] Wehr R , Saleska S R . The long-solved problem of the best-fit straight line: Application to isotopic mixing lines. Biogeosciences 2016, 14, 17-29, doi: doi:10.5194/bg-14-17-2017.

Round  2

Reviewer 1 Report

The revised manuscript has improved from previous version.  The authors have responded to all my comments. I only have a few more in responses:

1)   The Jaffe et al. (2005) address Asian mercury emissions by measurements at remote background location of Hedo and Mt. Bachelor. That’s not the same target scale as this study. For remote background locations, the upwind air masses are well-mixed and thus measurements at these locations can represent a large upwind region (the Asian emissions).  Their assumption on constant background is thus a fine 1st order approximation. But for this study, the measured location is very close to mainland China, and the goal is to estimate YRD emissions.  It is not valid to assume the air masses upwind the YRD are all well-mixed and constant. In reality, the upwind southeast or northeast provinces can contribute significantly to the measured signals at this site giving the long atmospheric lifetime of CO2 and CH4. Thus upwind of YRD can expect to be varying over time and space; the constant background assumption should not be applied here.  Without properly removing all upwind influneces, the calculated emission ratios cannot be good representative of the YRD region only, but a combination of YRD and further upwind areas. The authors have added a new estimate of these ratios using the 5th percentiles as background. The 5th percentile is a rough approximation that the authors need further reasoning. Why subtract the 5th percentile value from the mean instead of actual data points, and estimate the slope (as the ratio) using regression? I also recommend estimating the uncertainty in these ratios and resulting uncertainty in final emission rate due to the choice of background. The imperfect correlations between CO2 and CH4 in Figure 8c and 8d (R^2 < 0.8) could be a result of the non-constant backgrounds.

2)   Regarding to the calibration scale, I recommend the authors gather necessary references to the standard gases they used. Without a clear link to the current world standard (WMO2004x), it will devalue the use of this paper and the reported data in future works. 

3) For footprint analysis of sensitivity to surface emissions, do you analyze the result of footprint X emission per grid? Without the gridded surface emissions (e.g. from inventor), the footprint just gives an idea of when the particles are from but not the direct influence of surface emissions.

Author Response

Response to Reviewer 1 Comment

Point 1: The Jaffe et al. (2005) address Asian mercury emissions by measurements at remote background location of Hedo and Mt. Bachelor. That’s not the same target scale as this study. For remote background locations, the upwind air masses are well-mixed and thus measurements at these locations can represent a large upwind region (the Asian emissions).  Their assumption on constant background is thus a fine 1st order approximation. But for this study, the measured location is very close to mainland China, and the goal is to estimate YRD emissions.  It is not valid to assume the air masses upwind the YRD are all well-mixed and constant. In reality, the upwind southeast or northeast provinces can contribute significantly to the measured signals at this site giving the long atmospheric lifetime of CO2 and CH4. Thus upwind of YRD can expect to be varying over time and space; the constant background assumption should not be applied here.  Without properly removing all upwind influences, the calculated emission ratios cannot be good representative of the YRD region only, but a combination of YRD and further upwind areas. The authors have added a new estimate of these ratios using the 5th percentiles as background. The 5th percentile is a rough approximation that the authors need further reasoning. Why subtract the 5th percentile value from the mean instead of actual data points, and estimate the slope (as the ratio) using regression? I also recommend estimating the uncertainty in these ratios and resulting uncertainty in final emission rate due to the choice of background. The imperfect correlations between CO2 and CH4 in Figure 8c and 8d (R^2 < 0.8) could be a result of the non-constant backgrounds.

Response 1: It is true that in Jaffe et al. (2005) the background concentration is well defined and nearly constant. In our study, the background was not constant, thus leading to imperfect correlation between CO2 and CH4. It is worthy pointing out that our study is not the first one to apply the correlation analysis to obtain emissions ratio at the regional scale. For example, Wong et al. [1] used it to estimate the CH4:CO2 emissions ratio for the Los Angles airshed. Sigler et al. [2] used it to obtain the Hg:CO2 emissions ratio for Northeast US.

We agree with the reviewer that the imperfect correlations could be a result of the non-constant backgrounds. The uncertainty from this imperfect correlation is noted in Figure 12.

We used value 5th percentile value from our measurements as the background because the actual background was not known. If we remove this background from the measurement and then apply the linear regression, we obtain nearly identical slope value as the original GMR without removal of the background. We also tried the 10th percentile and the 15th percentile as the background values, and the results remained essentially the same (see figure below).   

Comparison of CH4/CO2 emissions ratio (ppm/ppm) estimated with different methods (The figure can be found in the attached doc file).

Point 2: Regarding to the calibration scale, I recommend the authors gather necessary references to the standard gases they used. Without a clear link to the current world standard (WMO2004x), it will devalue the use of this paper and the reported data in future works. 

Response 2: We have added the following text in the methods section:

“No standards wee available for us to trace our calibration gases to the WMO scale. NIM participated in two inter-agency comparison experiments on calibration standards including standards traceable to the WMO scale. The results can be found in refs [29, 30].” (Lines 136 -138)

Point 3: For footprint analysis of sensitivity to surface emissions, do you analyze the result of footprint X emission per grid? Without the gridded surface emissions (e.g. from inventor), the footprint just gives an idea of when the particles are from but not the direct influence of surface emissions.

Response 3: No, the analysis does not require the emission in per grid. The reviewer is correct that the footprint just gives an idea of the source region that impacts the measurement, but it does not give weighting factors to the emissions sources within the region. In response to this comment, we have added the following text:

“The footprint analysis revealed the source region mostly likely to have impacted the daytime measurement at MLW. The actual probability value, or weighting factor, was not used later when we aggregated the inventory emission data to the whole YRD region.” (Lines 218 -220)

Reference

[1] Wong, K.W.; Fu, D.; Pongetti, T.J; Newman, S.; Kort, E.A.; Duren, R.; Hsu, Y.K.; Miller, C.E.; Yung, Y.L.; Sander, S.P. Mapping CH4: CO2 ratios in Los Angeles with CLARS-FTS from Mount Wilson, California. Atmospheric Chemistry & Physics 2015, 15, 241–252, doi: 10.5194/acp-15-241-2015.

[2] Sigler, J.M.; Lee, X. Recent trends in anthropogenic mercury emission in the northeast United States. Journal of Geophysical Research Atmospheres 2006, 111, 3131–3148, doi: 10.1029/2005JD006814.
